# Modulation of Specific Sphingosine-1-Phosphate Receptors Augments a Repair Mediating Schwann Cell Phenotype

**DOI:** 10.3390/ijms231810311

**Published:** 2022-09-07

**Authors:** Jessica Schira-Heinen, Luzhou Wang, Seda Akgün, Sofia Blum, Brigida Ziegler, André Heinen, Hans-Peter Hartung, Patrick Küry

**Affiliations:** Neuroregeneration Laboratory, Department of Neurology, Medical Faculty, Heinrich-Heine-University, 40225 Düsseldorf, Germany

**Keywords:** dedifferentiation, glia, peripheral nerve regeneration, PNS, S1P, transdifferentiation

## Abstract

Transdifferentiation of Schwann cells is essential for functional peripheral nerve regeneration after injury. By activating a repair program, Schwann cells promote functional axonal regeneration and remyelination. However, chronic denervation, aging, metabolic diseases, or chronic inflammatory processes reduce the transdifferentiation capacity and thus diminish peripheral nerve repair. It was recently described that the sphingosine-1-phosphate receptor (S1PR) agonist Fingolimod enhances the Schwann cell repair phenotype by activation of dedifferentiation markers and concomitant release of trophic factors resulting in enhanced neurite growth. Since Fingolimod targets four out of five S1PRs (S1P1, S1P3-5) possibly leading to non-specific adverse effects, identification of the main receptor(s) responsible for the observed phenotypic changes is mandatory for future specific treatment approaches. Our experiments revealed that S1P3 dominates and that along with S1P1 acts as the responsible receptor for Schwann cell transdifferentiation as revealed by the combinatory application of specific agonists and antagonists. Targeting both receptors reduced the expression of myelin-associated genes, increased *PDGF-BB* representing enhanced trophic factor expression likely to result from c-Jun induction. Furthermore, we demonstrated that S1P4 and S1P5 play only a minor role in the adaptation of the repair phenotype. In conclusion, modulation of S1P1 and S1P3 could be effective to enhance the Schwann cell repair phenotype and thus stimulate proper nerve repair.

## 1. Introduction

Despite similarities, repair Schwann cells (SCs), also called Büngner cells, differ from immature Schwann cells by the activation of a repair program and downregulation of myelin gene expression (reviewed in [1]). Acute injuries to the peripheral nervous system (PNS) lead to de- or transdifferentiation of Schwann cells orchestrating peripheral nerve regeneration by initiation of the immune response, support of myelin clearance as well as axonal regeneration by providing trophic factors and axonal guidance molecules [2,3]. Although peripheral nerve regeneration is quite efficient compared to the injured central nervous system, the overall regeneration capacity diminishes over time as well as in response to pathological changes in the nervous tissue. For example, it was demonstrated that Schwann cells in chronically denervated nerves lose the capacity to transdifferentiate with subsequent failure of trophic factor secretion and nerve regeneration [4,5]. Moreover, aging reduces the regeneration capability due to less efficient activation of the transdifferentiation program [6] including lower amounts of trophic factor secretion (reviewed in [7]) and induction of chronic inflammatory processes [8]. Pharmacological modulation of Schwann cells to adopt and maintain the repair phenotype thus fulfills unmet clinical needs, particularly for the extended periods typically required for nerve repair in humans [9]. Thus, reactivation of Schwann cells in distal nerve stumps and stimulating their intrinsic capacity could be effective in order to enhance the nerve regeneration potential [5,10,11].

Sphingosine-1-phosphate (S1P) is derived from ceramide via deacylation by ceramidases resulting in the intermediate sphingosine which is phosphorylated by sphingosine kinases 1/2 [12]. S1P primarily acts as an extracellular signaling molecule via binding to five different G-protein-coupled receptors (S1P1-5) regulating, e.g., cell growth, proliferation, and migration as well as apoptosis and inflammation [13,14,15]. Each S1P receptor is able to trigger distinct signaling pathways and is therefore potentially implicated in regulatory functions of various cellular responses. S1P1 acts via activation of G-protein G_i_ whereas S1P2 and S1P3 act via G_i_, G_q_, and G_12/13_ activation, and S1P4 and S1P5 activate G_i_ and G_12/13_.

Modulation of S1P receptors by application of FTY720P (Fingolimod) to both juvenile and adult Schwann cells promotes the generation of the repair mediating Schwann cell phenotype characterized by a downregulation of myelin proteins and temporal activation of c-Jun, the latter of which is known to trigger Schwann cell transdifferentiation [16,17,18]. Moreover, mass spectrometry analysis revealed that FTY720P induces the secretion of a number of trophic factors involved in glial-mediated PNS repair via promoting neurite growth of dorsal root ganglion (DRG) explants on both growth permissive and growth inhibitory substrates [16,19]. Additionally, the TGFβ-Smad signaling pathway is activated in response to Fingolimod treatment [19]. TGFβ signaling was previously suggested to promote nerve regeneration by reactivation of a growth permissive Schwann cell phenotype [20,21,22]. Transdifferentiation processes after PNS injury seem to be similar to an epithelial mesenchymal transition (EMT) behavior [2,22]. Interestingly, it was shown that the EMT is modulated by S1P signaling in a TGFβ-dependent manner [23,24] indicating that S1P receptor (S1PR) modulation could be used as a therapeutic option to promote nerve regeneration via an engaged Schwann cell transdifferentiation process.

Since Fingolimod targets four out of five S1PRs (S1P1, S1P3-5) [25,26] possibly leading to unspecific side effects, identification of the main receptor responsible for SC transdifferentiation is mandatory for future specific treatment approaches. We, therefore, investigated which S1PR subtype dominated SC transdifferentiation and whether the specific modulation of single S1PR subtypes is sufficient to promote the generation of the Schwann cell repair phenotype.

## 2. Results

### 2.1. S1P Receptor Expression in Rat Schwann Cells

We aimed at identifying the S1P receptors involved in phenotypic changes representing repair Schwann cells. Whereas large differences in receptor subtype expression were initially reported in Schwann cell cultures [27], a later study described them to be expressed at similar levels in cultured neonatal rat Schwann cells [28]. Due to these inconsistencies, we reevaluated *S1P1-5* gene expression levels in neonatal rat Schwann cells cultured according to well-established protocols [19,29,30]. This demonstrated that compared to *S1P1*, *S1P4* and *S1P5* showed slightly lower abundances whereas *S1P2* transcript levels were >150-fold and *S1P3* transcript levels were >2000-fold increased, respectively, indicating that these two receptors are highly expressed in SCs (Figure 1). Given the low levels of S1P4 and S1P5 and since Fingolimod does not target S1P2, we focused on S1P1 and S1P3.

### 2.2. Effect of S1P1 Modulation on the Schwann Cell Phenotype

Treatment with Fingolimod induces a repair Schwann cell phenotype [16,19] which could be triggered by targeting all S1P receptors except S1P2. To investigate whether S1P1 is necessary for the induction of the SC repair phenotype, an S1P1 antagonist (EX26) was used in combination with Fingolimod, and expression patterns of corresponding markers were analyzed. Fingolimod induced the expression of *growth-associated protein 43* (*GAP43*, Figure 2A) and the *growth factor platelet-derived growth factor-beta* (*PDGF-BB*, Figure 2B) known to support axonal regeneration. Transition to repair cells was further accompanied by a reduced expression of transcripts encoding the *myelin and lymphocyte protein* (*Mal*, Figure 2C) as well as the myelination-associated transcription factors *early growth response 2* (*Egr2*; *Krox20*, Figure 2E) and *octamer transcription factor 6* (*Oct-6*, Figure 2F). Furthermore, *tyrosinase-related protein 1* (*Tyrp1*) expression, known to be controlled by c-Jun [17], was downregulated upon Fingolimod treatment (Figure 2D). Besides treatment with Fingolimod as a positive control, EX26 was applied alone to investigate possible side effects of the antagonist itself. Antagonizing S1P1 alone showed no effect on the SC phenotype since the expression of the aforementioned markers was unchanged with the exception of slightly increased *Tyrp1* levels (Figure 2D). Importantly, the combination of Fingolimod with EX26 led to a striking change in the expression pattern with Fingolimod-specific *GAP43*, *PDGF-BB*, and *Mal* responses (Figure 2A–C) being almost abolished. Nevertheless, *Krox20* expression after co-treatment with Fingolimod and EX26 did not differ significantly from EX26-only treated cells (Figure 2E), while *Tyrp1* (Figure 2D) and *Oct-6* (Figure 2F) were similarly expressed as in Fingolimod-only treated cells. These observations provide evidence that Schwann cell transdifferentiation boosted by Fingolimod is at least partially dependent on S1P1. This was further supported by the finding that c-Jun and p-c-Jun proteins, which are both further induced by Fingolimod [16], can be abolished by co-application of the S1P1 antagonist EX26 (Figure 2G,H,J,K) whereas EX26 alone did not have any impact on these protein abundances. Quantification of PTEN and p-PTEN protein abundances revealed no difference after Fingolimod treatment reflecting previous results [16]. Additionally, levels of PTEN and p-PTEN were not affected by antagonizing S1P1 (Figure 2G,I,J,L).

To further investigate whether S1P1 is important for the phenotypic transition of SCs into repair cells, the selective S1P1 receptor agonist Ponesimod was used to target S1P1. Gene expression analysis revealed that stimulation of S1P1 by Ponesimod led to a significant induction of the regeneration-associated genes *GAP43* (Figure 3A) and *PDGF-BB* (Figure 3B). In addition, a significant decline in expression of myelin-associated genes *Mal* (Figure 3C), *Tyrp1* (Figure 3D), *Krox20* (Figure 3E) and *Oct-6* (Figure 3F) was observed. Upon combination with EX26, *GAP43*, *PDGF-BB*, and *Mal* gene expression changes were almost neutralized and similar to controls. Compared to treatment with Ponesimod alone, the combination with EX26 led to a significantly increased expression of *Tyrp1*, *Krox20*, and *Oct-6* which was, however, still not reaching levels of control SCs. By applying Ponesimod we confirmed an implication of S1P1 in the transition of SCs to the repair phenotype.

### 2.3. Role of S1P3 in Schwann Cell Transdifferentiation

The high expression level of *S1P3* (Figure 1) prompted investigations on its additional role in SC transdifferentiation. For this purpose, an S1P3 antagonist (TY52156) was co-applied with Fingolimod and subsequently, expression levels of the aforementioned repair SC marker genes were measured. Treatment with this S1P3 antagonist alone did not affect overall gene expression levels with exception of slightly induced *Mal* (Figure 4C) and *Krox20* expression levels (Figure 4E). The significant induction of *GAP43* and *PDGF-BB* in response to Fingolimod was completely abolished in presence of TY52156 (Figure 4A,B). Furthermore, Fingolimod-dependent downregulation of *Mal* (Figure 4C), *Tyrp1* (Figure 4D), *Krox20* (Figure 4E) and *Oct-6* (Figure 4F) transcript levels were successfully antagonized in presence of TY52156, with some gene expression levels even slightly exceeding those of control-treated cells. Likewise, induction of c-Jun protein expression by Fingolimod was reduced in the presence of TY52156, whereas PTEN and p-PTEN levels remained unchanged (Figure 4G–L). Thus, these results indicate that besides S1P1 also S1P3 enhances transdifferentiation into the repair SC phenotype. Hence, by applying this S1P3-specific antagonist the effect of Fingolimod on the generation of the repair mediating Schwann cell phenotype could be reversed.

In order to verify that S1P3 is involved in the induction of the repair mediating Schwann cell phenotype, an agonist specifically targeting S1P3 (CYM5541) was applied in subsequent stimulation experiments. Expression of *GAP43* and *PDGF-BB* was upregulated (Figure 5A,B) and myelin-related genes were downregulated (Figure 5C–F) in presence of CYM5541 reflecting the Fingolimod related expression patterns. Co-application of the S1P3 antagonist TY52156 completely abolished these transcriptional responses and even significantly boosted *Tyrp1* expression levels thus corroborating that S1P3 is essential to induce the repair of SC phenotype.

### 2.4. S1P4 and S1P5 Only Play a Minor Role in Schwann Cell Transdifferentiation

Due to the low expression levels of *S1P4* and *S1P5* in Schwann cells, we hypothesized that they exert only minor roles regarding transdifferentiation. To this end, we used specific agonists for S1P4 (CYM50260) and S1P5 (A971432) and subsequently performed gene expression analyses. As these molecules and their dosing are not well described we first tested a small variety of concentrations. Modulation of S1P4 resulted in concentration-dependent expression changes of *GAP43*, *PDGF-BB*, and *Tyrp1* whereas the myelin-associated genes *Mal*, *Krox20*, and *Oct-6* were not affected (Figure 6).

Since some genes were indeed regulated by this S1P4 agonist, we next used an S1P4 antagonist CYM50358 (S1P4 A) in combination with Fingolimod and again analyzed SC markers by qRT-PCR. This revealed that the S1P4 antagonist alone had no effect on gene expression and that this molecule was also not able to modulate Fingolimod-dependent responses (Figure 7).

Strikingly, independent of the concentration, modulation of S1P5 by means of S1P5 agonist A971432 application did not affect the expression of the repair SC markers (Figure 8). Thus, our data revealed that enhancement of the repair program gene expression by fingolimod did not require activation of S1P4 or S1P5 receptors.

## 3. Discussion

Transdifferentiation of myelinating and non-myelinating Schwann cells into a repair mediating cell type is an important prerequisite for successful nerve repair after peripheral nerve injury. Due to extended regeneration periods in humans, efficient transdifferentiation faces limitations. In addition, peripheral nerve pathologies induced by, e.g., inflammation, diabetes, or drugs reduce the nerve repair capacity resulting in life-long nerve impairment and consequently diminished quality of life. Thus, particularly in conditions of impeded nerve regeneration, pharmacological treatment to reactivate Schwann cells to adopt and maintain the repair phenotype could be effective in order to enhance nerve restoration.

S1P receptor modulation by Fingolimod treatment was shown to boost the repair phenotype of cultured neonatal and adult rat Schwann cells characterized by downregulation of myelin-associated genes and induction of trophic factor secretion [16,19]. However, to prevent adverse side effects, identification of the main S1P receptor responsible for transdifferentiation is mandatory to develop new therapeutic approaches to foster the SC repair phenotype and thus functional nerve repair. We here investigated the impact of S1P receptor modulators on the Schwann cell phenotype and demonstrated that S1P1 and S1P3 are the main contributors to Schwann cell transdifferentiation. Targeting both receptors reduced the expression of myelin-associated genes, and increased *PDGF-BB* representing enhanced trophic factor expression, both likely due to the induction of c-Jun.

It was already demonstrated that rat as well as human Schwann cells express S1P1 protein in vitro and in vivo [28] as well as other S1P receptors [27]. Nevertheless, we performed another qRT-PCR analysis to verify that Schwann cells which were prepared and cultured in a different manner also expressed *S1P1* (and other S1P receptors). Gene expression analysis revealed that *S1P3* is highly expressed in cultured Schwann cells compared to other S1P receptors (Figure 1). Thus, *S1P3* is likely to be the main Fingolimod target in SCs. Furthermore, *S1P2* is highly expressed in Schwann cells, however, since Fingolimod does not target S1P2, this receptor subtype was excluded from further investigation. Nevertheless, the involvement of S1P2 in chemotherapy-induced neuropathy [31] and its high expression levels in Schwann cells suggest an important role in the PNS and should be addressed in future investigations. In comparison, *S1P4* and *S1P5* relative expression levels were low which is in agreement with previous observations [27]. Of note, RNA-sequencing of SCs purified from sciatic nerves after injury revealed low expression levels of *S1P5* [22] indicating that our observations in cultured Schwann cells reflect physiological *S1P5* expression levels in vivo. Interestingly, dedifferentiated Schwann cells derived from the bridge and from the distal nerve part significantly downregulated *S1P3* whereas the other S1P receptors were not affected by the injury [22]. It thus appears that S1P3 has a physiological relevance during the nerve repair process which needs further characterization. At protein level, it is known that S1P1 is internalized after binding to Fingolimod, hence protein turnover and localization could be highly dynamic depending on extrinsic signals. Therefore, future experiments have to address the temporal and spatial S1P1 and S1P3 localization as well as posttranscriptional/-translational regulation in cultured Schwann cells and in vivo after nerve injury.

By applying combinations of Fingolimod with antagonists for S1P1 and S1P3, respectively, we provided evidence that both receptors are involved in Schwann cell transdifferentiation. Fingolimod enhanced the SC repair phenotype characterized by changes in transcription factor expression including a downregulation of *Krox20* and *Oct-6* accompanied by an induction of *GAP43* and *PDGF-BB* as well as higher c-Jun protein levels, the latter of which is known to be a key regulator in Schwann cell transdifferentiation [17]. Similar to Fingolimod treatment, S1P1 antagonism by EX26 reverses the pathology of experimental autoimmune encephalomyelitis (EAE) which depends on significant lymphocyte sequestration in the lymph nodes [32]. In Schwann cells, however, S1P1 antagonism by EX26 application revealed no significant impact on the cellular phenotype (Figure 2). Since Fingolimod promotes the internalization of S1P1 which is counteracted by EX26 [32] it appears that S1P1 internalization could indeed be relevant for Schwann cell transdifferentiation.

Combined application of Fingolimod with EX26 (Figure 2) partially abolished the Fingolimod-mediated phenotypic changes. Co-application of EX26 neutralized the Fingolimod-mediated upregulation of the pro-regenerative factors *GAP43* and *PDGF-BB*, accompanied by reduced expression of *Mal*. However, expression levels of *Tyrp1*, *Krox20*, and *Oct-6* were similar to Fingolimod-treated cells. Likewise, the S1P1 agonist Ponesimod enhanced the repair phenotype, however, EX26 antagonism was only partially effective to neutralize the phenotypic changes (Figure 3). In contrast, Fingolimod-mediated repair phenotype adaption seems to be more effectively neutralized by the S1P3 antagonist TY52156 than EX26 since the Fingolimod-mediated phenotypic changes were completely abolished (Figure 4). The S1P3 antagonist TY52156 was shown to inhibit Fingolimod-induced bradycardia in vivo [33] which is a well-known side effect of Fingolimod treatment. Oral bioavailability makes TY52156 attractive for future therapeutic approaches, however, needs further validation [34]. Interestingly, targeting S1P3 by TY52156 application slightly induced *Mal* and *Krox20* expression, which is opposite to the effect of Fingolimod application. It thus appears that S1P3 plays an essential role in Schwann cell phenotypic adaptations to Fingolimod. By using the S1P3 agonist CYM5541, the Schwann cell repair phenotype was successfully induced (Figure 5), thus suggesting that S1P3 may have a role in SC transdifferentiation. CYM5541 was stated to specifically bind to S1P3 when applied in concentrations up to 10 µM. Furthermore, this S1P3 agonist is supposed to occupy a differential chemical space in the ligand binding pocket than S1P. Thus, this allosteric binding site may account for the high S1P3 selectivity of CYM5541 [35], especially in low concentrations as used in the presented study. The repair phenotype effect of CYM5541 was completely neutralized by co-application with the S1P3 antagonist TY52156, thus confirming the important role of S1P3 in Schwann cell transdifferentiation. The induction of higher c-Jun as well as p-c-Jun levels by Fingolimod was likewise neutralized by co-application of S1P1 and S1P3 antagonists, respectively (Figure 2 and Figure 4). Both receptors are known to target a wide spectrum of downstream effectors. Whereas S1P1 selectively binds to G_αi/0_, S1P3 acts via G_αi/0_, G_αq,_ and G_12/13_, the latter of which is known to regulate Jun kinase (JNK) which in turn phosphorylates c-Jun (reviewed by [36]). Crosstalk between both S1P receptor-activated pathways could explain that antagonism of both S1P receptors leads to neutralization of the Fingolimod-induced increase of p-c-Jun levels. Future experiments should address a detailed pathway analysis with a focus on S1P1 and S1P3-based induction of c-Jun signaling and putative crosstalk of both receptors.

Expression levels of *S1P4* and *S1P5* were relatively low in cultured Schwann cells (Figure 1). However, since Fingolimod targets both receptors we investigated their role in Schwann cell transdifferentiation. Targeting S1P4 by the agonist CYM50260 revealed a concentration-dependent effect on the expression levels of *GAP43*, *PDGF-BB*, and *Tyrp1* of which *GAP43* and *Tyrp1* were contrarily regulated compared to Fingolimod-induced phenotypic changes (Figure 6). We can only speculate on the activity of this supposed S1P4 agonist in that its specificity is not that tight. On the other hand, despite evidence that S1P4 does not appear to be one of the main receptors for Schwann cells, it nevertheless might interfere with S1P1 and S1P3. Additionally, antagonism of S1P4 revealed no effect on the SC repair marker expression (Figure 7) indicating that this receptor only plays a minor role in the adaptation of the repair phenotype. Similarly, targeting of S1P5 revealed no phenotypic changes (Figure 8). Based on our observations we therefore conclude that S1P3 modulation alone could be sufficient to efficiently support a SC repair phenotype and might probably come with fewer side effects.

As previously shown, cultured neonatal and adult Schwann cells behave similarly regarding phenotypic changes after Fingolimod treatment, therefore, only neonatal Schwann cells were used in the present study. Schwann cell reactions upon S1P modulation have so far not been studied in vivo. Upcoming experiments will therefore have to functionally investigate the role of S1P1 and S1P3 during peripheral nerve development and after peripheral nerve injury, probably by Schwann cell-specific ablation of each S1P receptor. This is of high relevance since, in antagonist/agonist combinatory experiments, binding competition of the molecules could not fully be excluded.

In addition, since targeting S1P3 appears to be an attractive treatment option to promote nerve repair, the impact of a (local or systemic) application of specific agonists must be addressed in peripheral nerve injury models. To our knowledge, the role of the S1PRs-S1P axis in Schwann cells after peripheral nerve injury has not been investigated so far. In future studies, molecular mechanisms as well as the functionality of S1PR in PNS regeneration should be addressed. Furthermore, as no detailed analysis of S1P itself in peripheral nerves after injury in vivo has been performed, future investigations will most likely also include lipidomics analyses. Interestingly, a recent study revealed that adoption of the repair phenotype in mice is accompanied by changes in the lipid metabolism including decreased levels of S1P which, however, is delayed in human injured nerves [37]. These observations suggest that pharmacological treatment of S1P receptors could lead to different effects in human Schwann cells. Thus, the transfer of S1PR modulation to human Schwann cells derived from, e.g., nerve biopsies or differentiated from induced pluripotent stem cells represents yet another important step toward developing new treatment options.

## 4. Materials and Methods

### 4.1. Primary Neonatal Schwann Cell Culture

Primary neonatal rat Schwann cells were prepared as described previously [29,30]. Briefly, sciatic nerves were obtained from ten rat pups (postnatal days 0 to 2), the surrounding epineurium was stripped off and blood vessels were removed. Nerves were cut into pieces of approximately 1–2 mm in length and incubated with a trypsin/collagenase mixture (2.5%/0.6%, respectively, both Sigma-Aldrich, Taufkirchen, Germany) in DMEM (Gibco, Life Technologies GmbH, Darmstadt, Germany) for 1 h at 37 °C. After stopping the digestion process, cell homogenate was filtered using a 60 µm gaze filter and transferred into non-coated cell culture flasks. In order to remove contaminating fibroblasts, cells were treated twice with 10 µM arabinosyl-cytosine (Ara-C, Sigma-Aldrich) followed by complement lysis using anti-Thy1.1 antibodies (Bio-Rad Cat# MCA04G, RRID:AB_322809) and baby rabbit complement (Cedarlane, Burlington, ON, Canada), resulting in purified Schwann cell cultures. To promote Schwann cell proliferation, 100 µg/mL bovine pituitary extract (Merck Millipore, Darmstadt, Germany) was applied after complement lysis. The purity of cultures was assessed by staining using an anti-S100 antibody (Sigma-Aldrich Cat# S2644, RRID:AB_477501). Afterwards, neonatal Schwann cells were cultured in poly-D-lysine (PDL, 0.1 mg/mL, Sigma-Aldrich) coated cell culture flasks at 37 °C and 10% CO_2_ in DMEM supplemented with 10% FBS (Lonza, Cologne, Germany), 2 mM L-glutamine (Thermo Fisher Scientific, Darmstadt, Germany) and penicillin/streptomycin (Thermo Fisher Scientific) in presence of 2 µM forskolin (Sigma-Aldrich). Schwann cell medium was exchanged every second day.

### 4.2. Modulation of S1P Receptors

For treatment with S1PR modulators, Schwann cells were seeded at a density of 25,000–35,000 Schwann cells/cm^2^ on PDL-coated 24-well plates. On day 2 after seeding, Schwann cells were incubated in a Schwann cell medium together with S1PR modulators for another 48 h. Fingolimod (0.1 µM FTY720P, Echelon Biosciences, Salt Lake City, UT, USA) was always used as a positive control. In addition, S1PR agonists were used to specifically modulate receptors S1P1 as well as S1P3-5. All substances were dissolved in DMSO (Sigma-Aldrich) unless otherwise stated: 7.5 µM Ponesimod (S1P1 agonist, Selleckchem, Houston, TX, USA); 15 µM EX26 (S1P1 antagonist, Tocris Bioscience, Bristol, UK); 5 µM CYM5541 (S1P3 agonist, Tocris); 5 µM TY52156 (S1P3 antagonist, Tocris), 1.25 µM, 2.5 µM, 5 µM and 10 µM CYM50260 (S1P4 agonist, Tocris); 1 µM CYM50358 hydrochloride (S1P4 antagonist, Tocris); 1.25 µM, 2.5 µM, 5 µM, and 10 µM A971432 (S1P5 agonist, Tocris, solved in 1N HCl). For each treatment condition, 3–4 wells of a 24-well plate were pooled and used for quantitative real-time-PCR (qRT-PCR) analysis.

### 4.3. Quantitative Gene Expression Analysis

After stimulation with S1PR modulators for 48 h, Schwann cells were washed once with PBS and lysed with RLT buffer (Qiagen, Hilden, Germany). RNA isolation was performed by using the RNeasy Mini Kit (Qiagen) according to the manufacturer’s instructions including DNase digestion (RNase free DNase Kit, Qiagen) to avoid genomic DNA contamination. RNA concentrations were measured by NanoDrop ND-1000 spectrophotometer. Only samples with high-quality RNA (260/280 > 1.8) were further used for qRT-PCR analysis. Synthesis of cDNA was performed by means of the High Capacity cDNA Reverse Transcription Kit with RNase inhibitor (Life Technologies) using 250 ng total RNA per reaction. Quantitative determination of relative gene expression was performed on Applied Biosystems 7900 HT Fast Real-Time PCR System using primers listed in Table 1 and SYBR Green master mix (Life Technologies). Data were analyzed according to the manufacturer’s ∆∆Ct method (Applied Biosystems). *ODC* and *GAPDH* were used as reference genes. Each sample was related to the corresponding buffer-treated control sample.

### 4.4. Western Blot Analysis

For protein analysis, 1.5–2 × 10^6^ Schwann cells were seeded on PDL-coated 10 cm dishes. On day 2 after seeding, cells were incubated in Schwann cell medium together with S1PR modulators for another 24 h using the same concentrations as for gene expression analysis. Prior to cell harvest, the medium was removed, and cells were washed 3× with ice-cold PBS to prevent protein contaminants originating from the FBS-containing culture medium. Afterward, 2 mL PBS was added to the dish, and cells were detached from the dish using a cell scraper (Sigma Aldrich). The cell suspension was centrifuged for 10 min at 2100 rpm and 4 °C. The supernatant was removed, and cell pellets were immediately frozen and stored at −80 °C. Cells were lysed in RIPA buffer (Cell Signaling Technology, Danvers, MA, USA) supplemented with HALT^TM^ Protease/Phosphatase inhibitor cocktail and EDTA (both Thermo Fisher Scientific). Lysates were sonicated (SonopulsHD2070; 50% power, pulse 0.5 s on and 0.5 s off) and centrifuged for 10 min at 14,000 rpm and 4 °C. Protein concentrations of supernatants were determined by DC Protein assay (BioRad, Cat# 5000112). Same protein amounts for each sample (30 µg to detect non-phosphorylated proteins or 50 µg to detect phosphorylated proteins) were diluted in Bolt LDS sample buffer supplemented with Bolt sample reducing agent, incubated at 70 °C for 10 min, loaded onto an SDS-PAGE (Bolt 12% Bis-Tris Plus Gel) and run for 15 min at 50 V followed by 200 V for 45 min (Sure Lock X Cell, all Thermo Fisher Scientific). Proteins were blotted on nitrocellulose membranes pre-incubated with blot-buffer by using the Power Blotter System (all Thermo Fisher Scientific). Afterward, proteins were visualized by Pierce^TM^ Reversible Protein Stain Kit and documented by using Fusion SL Vilber Lourmat imaging system (Vilber Lourmat, Eberhardzell, Germany) and subsequently blocked with 1% milk powder in TBS for 1 h at RT. Primary antibodies were diluted in blocking buffer with following concentrations: c-Jun (Cell Signaling Technology Cat# 9165, RRID:AB_2130165) 1:2000, p-c-Jun (Cell Signaling Technology Cat# 3270, RRID:AB_2129575) 1:1500, PTEN (Cell Signaling Technology Cat# 9188, RRID:AB_2253290) 1:2000, p-PTEN (Cell Signaling Technology Cat# 9551, RRID:AB_331407) 1:3000. After incubation overnight at 4 °C, membranes were washed with TBS/0.05% Tween 20 (3× 10 min) and the secondary antibody anti-rabbit IgG, HRP-linked (1:2000 in TBS/0.1% Tween20 supplemented with 1% milk powder, Cell Signaling Technology Cat# 7074, RRID:AB_2099233) was applied for 1 h at RT. After washing 3× with TBS/0.05 % Tween20, membranes were incubated with Super Signal West Pico Chemiluminescent Substrate (Thermo Fisher Scientific, Cat# 34579) for 5 min and signals were detected and analyzed by using the Fusion SL Vilber Lourmat imaging system including only unsaturated protein bands. Since most of the housekeeping proteins, such as GAPDH or actin, are highly abundant, detection signals are mostly out of the linear range which makes reliable quantification impossible. To prevent this technical issue, we here used total protein normalization, which is based on normalization of the abundance of target proteins to the total amount of protein in each lane. This method is independent of a single loading control and is recognized as the gold standard for quantitative Western Blot analysis [38,39]. Three individual experiments were performed (= 3 biological replicates). For two biological replicates, blots were performed twice (technical replicates).

### 4.5. Statistical Analysis

All data are shown as mean values and standard error of the mean (SEM). For statistical analysis and graph illustrations, Microsoft Excel and Graphpad Prism (version 8.0, San Diego, CA, RRID: SCR_002798) were used. For all shown data, significances were analyzed by using a one-way ANOVA with Dunnett’s multiple comparison test or Tukey’s multiple comparison test. Data were considered statistically significant at * *p* < 0.05, ** *p* < 0.01, *** *p* < 0.001. In order to ensure the clarity of the data, we refrained from labeling data that were not significantly different. n represents the number of independent experiments.

## Figures and Tables

**Figure 1 ijms-23-10311-f001:**
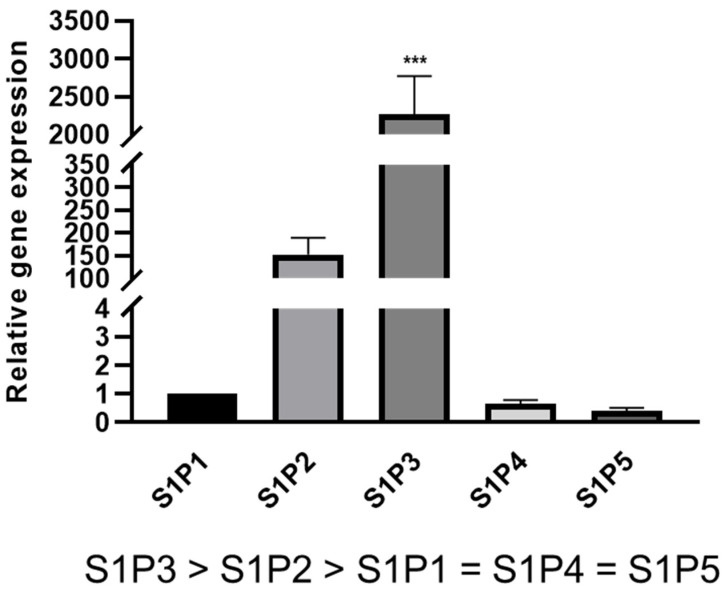
S1P receptor expression in cultured rat neonatal Schwann cells. Gene expression analysis revealed that *S1P2* and *S1P3* are highly expressed in SCs whereas *S1P4* and *S1P5* were slightly lower abundant compared to *S1P1*. For statistical analysis, a one-way ANOVA with Dunnett’s multiple comparison test was used, *** *p* < 0.001 (*n* = 5–6). Shown are the mean values ± SEM.

**Figure 2 ijms-23-10311-f002:**
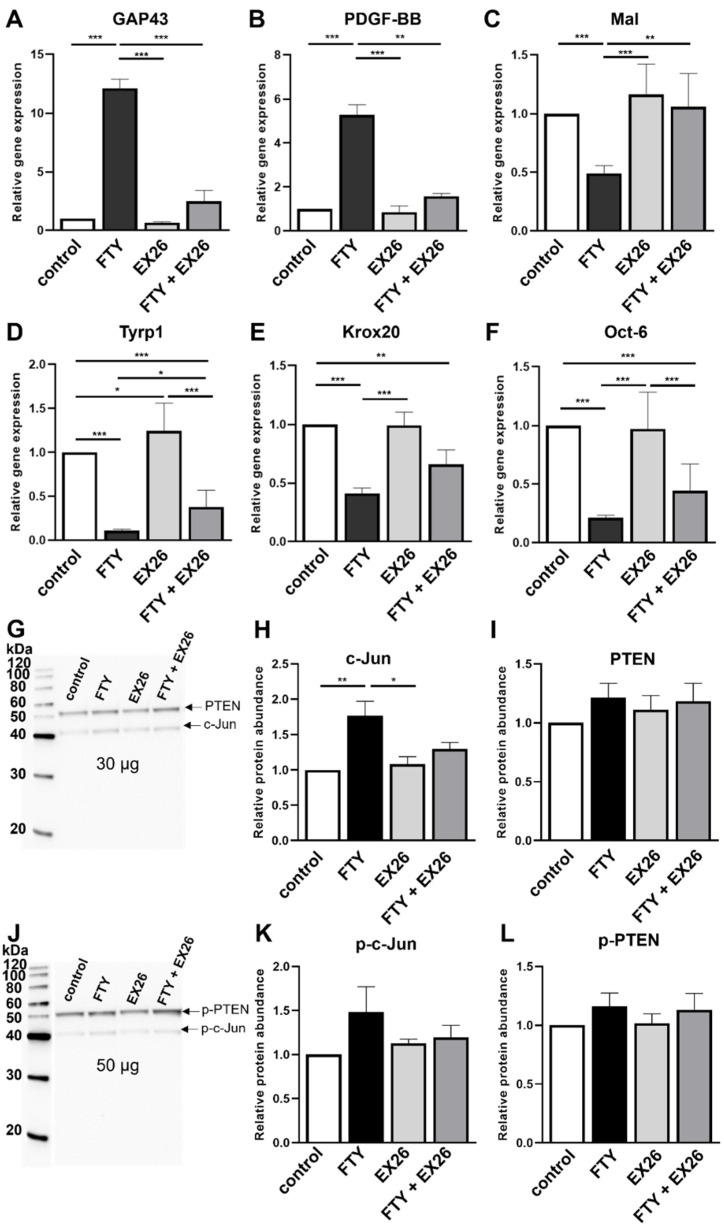
Antagonizing S1P1 reverses the effect of Fingolimod. Compared to Fingolimod (FTY) alone, the combination with the S1P1 antagonist EX26 abolished the transition to the repair phenotype. Upon combinatory treatment, *GAP43* (**A**), *PDGF-BB* (**B**) and *Mal* (**C**) showed similar expression levels compared to control (buffer) treated SCs. Expression levels of *Tyrp1* (**D**), *Krox20* (**E**) and *Oct-6* (**F**) slightly increased compared to Fingolimod. To exclude side effects of the antagonist, EX26 was applied alone revealing only a slight upregulation of *Tyrp1* compared to control. For statistical analysis, a one-way ANOVA with Tukey’s multiple comparison test was used, * *p* < 0.05, ** *p* < 0.01, *** *p* < 0.001. Shown are the mean values ± SEM with at least *n* = 4 biological replicates. Western Blot analysis (**G**,**J**) and respective quantification revealed an induction of c-Jun (**H**) and p-c-Jun (**K**) after Fingolimod treatment for 24h which was abolished by co-application with the S1P1 antagonist EX26. No effect on PTEN (**I**) and p-PTEN (**L**) protein abundances was observed, respectively. Representative Western Blots are shown (**G**,**K**). Quantitative data include mean values ± SEM of three biological replicates.

**Figure 3 ijms-23-10311-f003:**
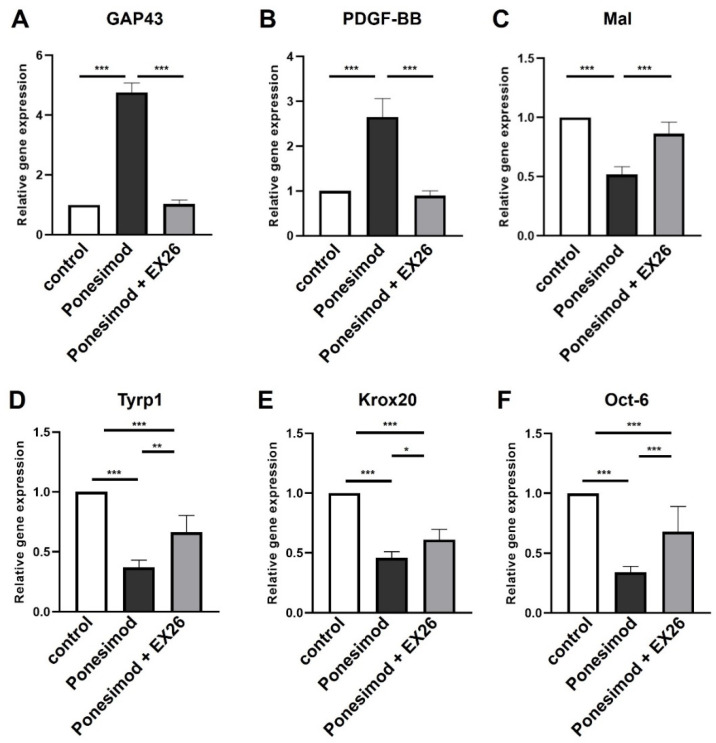
Ponesimod enhances the generation of a SC repair phenotype. qRT-PCR-based quantification of gene expression upon stimulation with the selective S1P1 agonist Ponesimod revealed a significant induction of *GAP43* (**A**) and *PDGF-BB* (**B**) expression whereas *Mal* (**C**), *Typr1* (**D**), *Krox20* (**E**) and *Oct-6* (**F**) expression were reduced. The combinatory treatment with EX26 almost abolished all Ponesimod-dependent effects. For statistical analysis, a one-way ANOVA with Tukey’s multiple comparison test was used, * *p* < 0.05, ** *p* < 0.01, *** *p* < 0.001 with at least *n* = 4 biological replicates. Shown are the mean values ± SEM.

**Figure 4 ijms-23-10311-f004:**
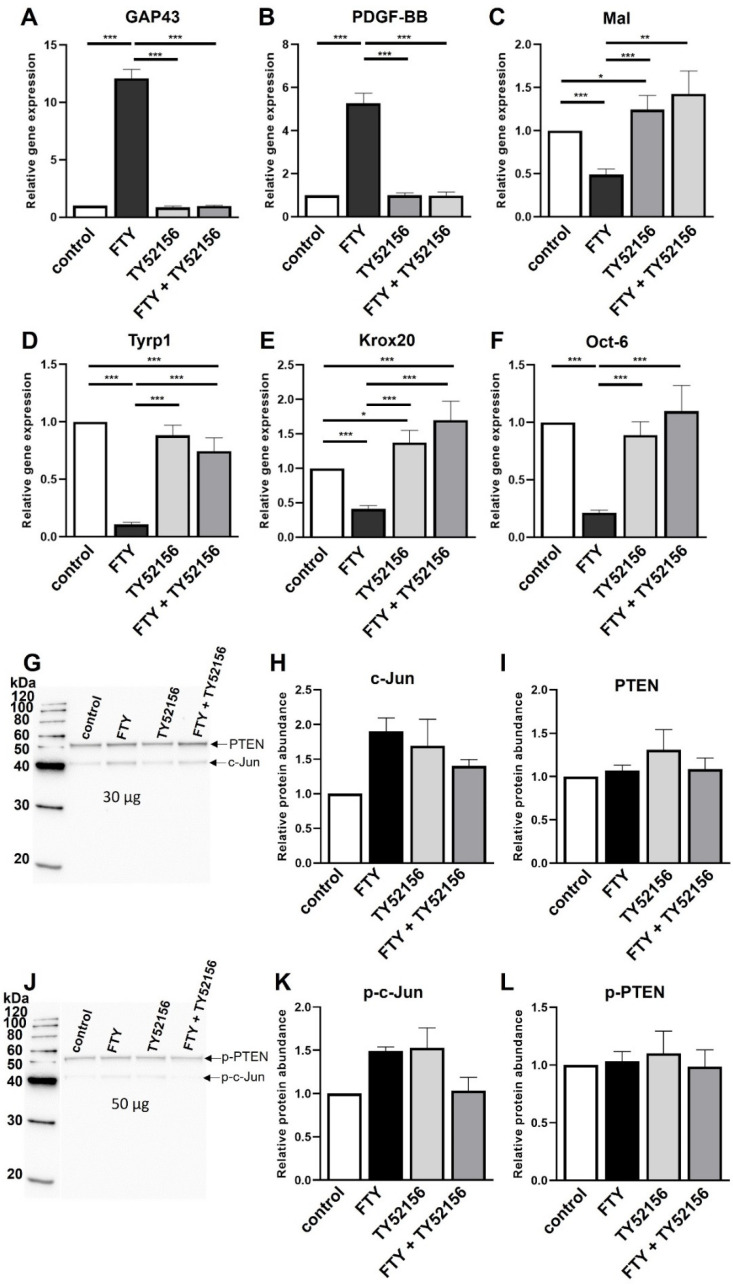
Induction of the repair Schwann cell phenotype by Fingolimod is neutralized by antagonizing S1P3. Co-application of Fingolimod (FTY) with TY52156 led to a reversed expression of *GAP43* (**A**), *PDGF-BB* (**B**), *Mal* (**C**), *Tyrp1* (**D**), *Krox20* (**E**) and *Oct-6* (**F**) compared to Fingolimod treatment alone having similar expression levels as control treated SCs. Treatment of SCs with the S1P3 antagonist alone led to a slight increase of *Mal* (**C**) and *Krox20* (**E**). For statistical analysis, a one-way ANOVA with Tukey’s multiple comparison test was used, * *p* < 0.05, ** *p* < 0.01, *** *p* < 0.001 with at least *n* = 4 biological replicates. Shown are the mean values ± SEM. Increased levels of c-Jun (**G**,**H**) and p-c-Jun (**J**,**K**) proteins induced by Fingolimod were abolished by co-application of the S1P3 antagonist whereas no effect on PTEN (**G**,**I**) and p-PTEN (**K**,**L**) protein levels was detected. Representative Western Blots are shown (**G**,**K**). Quantitative data include mean values ± SEM of three biological replicates.

**Figure 5 ijms-23-10311-f005:**
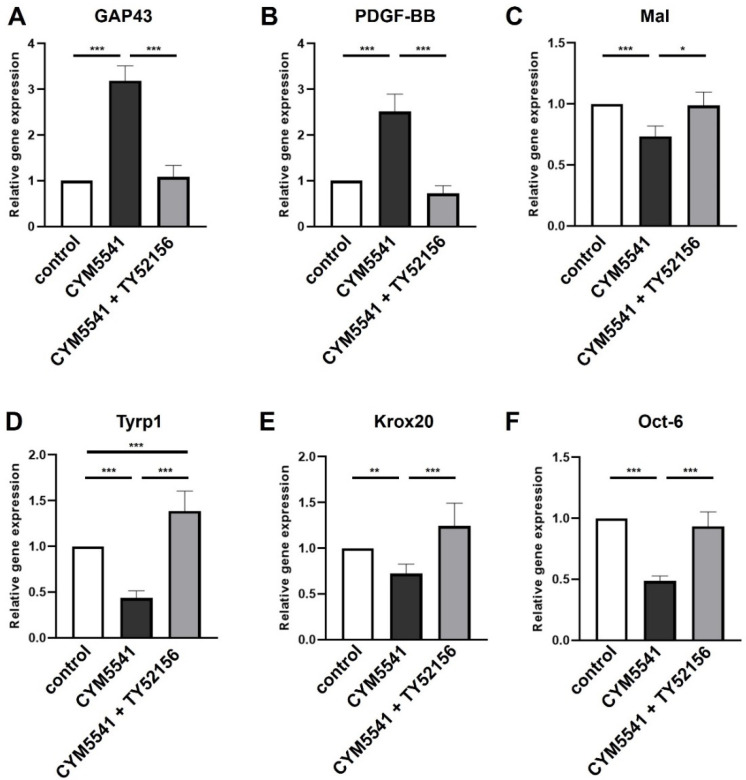
Specific stimulation of S1P3 enhances the Schwann cell repair phenotype. Application of the S1P3 agonist CYM5541 enhanced the SC repair phenotype including a significant induction of *GAP43* (**A**) and *PDGF*-BB (**B**) expression whereas the myelin-associated genes *Mal* (**C**), *Tyrp1* (**D**), *Krox20* (**E**) and *Oct-6* (**F**) were reduced. The effect of the S1P3 agonist was completely neutralized by co-application of the S1P3 antagonist TY52156. For statistical analysis, a one-way ANOVA with Tukey’s multiple comparison test was used, * *p* < 0.05, ** *p* < 0.01, *** *p* < 0.001 with at least *n* = 4 biological replicates.

**Figure 6 ijms-23-10311-f006:**
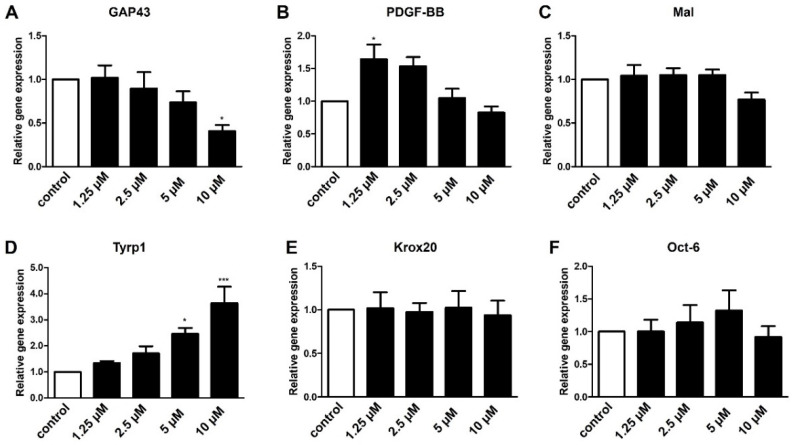
S1P4 plays only a minor role in Schwann cell transdifferentiation. Incubation with an agonist targeting S1P4 (CYM50260) led to a concentration-dependent downregulation of *GAP43* (**A**) and an induction of *PDGF-BB* (**B**) and *Tyrp1* (**D**) whereas *Mal* (**C**), *Krox20* (**E**) and *Oct-6* (**F**) were not affected by S1P4 modulation. For statistical analysis, a one-way ANOVA with Dunnett’s multiple comparison test was used, * *p* < 0.05, *** *p* < 0.001 (*n* = 3–4). Shown are the mean values ± SEM.

**Figure 7 ijms-23-10311-f007:**
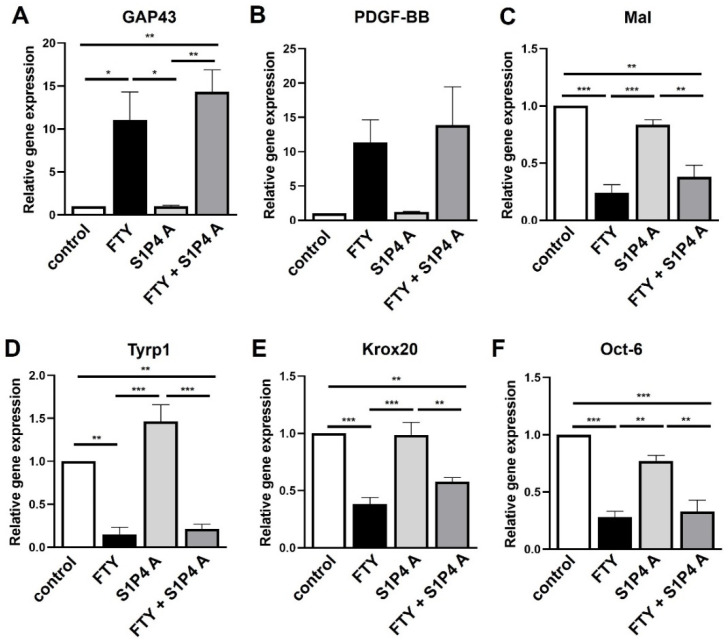
Neutralization of the S1P4 receptor does not affect the Schwann cell phenotype. qRT-PCR based quantification of gene expression upon application of an S1P4 antagonist (S1P4 A, CYM50358) alone as well as in combination with Fingolimod (FTY + S1P4 A) revealed no effect on the expression of *GAP43* (**A**), *PDGF-BB* (**B**), *Mal* (**C**), *Tyrp1* (**D**), *Krox20* (**E**) and *Oct-6* (**F**) whereas Fingolimod alone led to the typical expression pattern of all markers. For statistical analysis, a one-way ANOVA with Tukey’s multiple comparison test was used, * *p* < 0.05, ** *p* < 0.01, *** *p* < 0.001 (*n* = 3).

**Figure 8 ijms-23-10311-f008:**
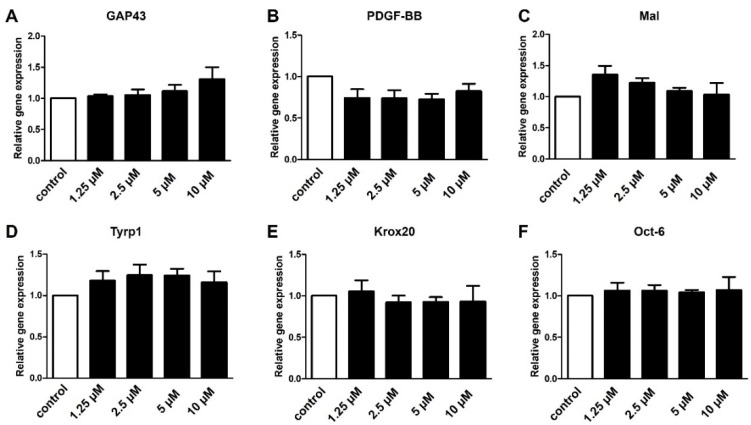
Schwann cell transdifferentiation does not depend on S1P5. Modulation of S1P5 with a specific agonist (A971432) applied in different concentrations did not affect the expression of *GAP43* (**A**), *PDGF-BB* (**B**), *Mal* (**C**), *Tyrp1* (**D**), *Krox20* (**E**) or *Oct-6* (**F**). For statistical analysis, a one-way ANOVA with Dunnett’s multiple comparison test was used, *n* = 4–5. Shown are the mean values ± SEM.

**Table 1 ijms-23-10311-t001:** Primer sequences.

Gene	Primer Sequence
*GAP43*	fwd: CCG GAG GAT AAG GCT CAT AAG G rev: TTG TTA TGT GTC CAC GGA AGC T
*GAPDH*	fwd: GAA CGG GAA GCT CAC TGG C rev: GCA TGT CAG ATC CAC AAC GG
*Krox20*	fwd: TTT TTC CAT CTC CGT GCC A rev: GAA CGG CTT TCG ATC AGG G
*Mal*	fwd: AGG AGG CCT TTG GTT ATC CC rev: GCA AAT GGC AGA TTT GGG TAC
*Oct-6*	fwd: GGC ACC CTC TAC GGT AAT GTG T rev: TTG AGC AGC GGT TTG AGC T
*ODC*	fwd: GGT TCC AGA GGC CAA ACA TC rev: GTT GCC ACA TTG ACC GTG AC
*PDGF-BB*	fwd: GTT CGG ACG GTG CGA ATC rev: GTG TGC TTA AAC TTT CGG TGC TT
*S1P1*	fwd: AAA TGC CCC AAC GGA GAC T rev: TCC ATG CCC GGG ATG AT
*S1P2*	fwd: GGT CAA GCT CTA CGG CAG TGA rev: AAG AGG CCC CAA TGA GCA T
*S1P3*	fwd: TCC TCA ACT CGG CCA TGA A rev: ACG CCG CAT CTC TTT GCT
*S1P4*	fwd: TGA GGC CCA GGG ACA GTT T rev: GGC TCT CGC ATC TTG AAG CT
*S1P5*	fwd: TTG GCT GTG TGC GCT TTC rev: GGT GCA TGG AAG CGA GGA T
*Tyrp1*	fwd: CTT CGT CAG GGC CTT GGA rev: GCA ATG ACA AAT TGA GGG TGA GT

## Data Availability

Not applicable.

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
