# Peer review of "Modulation of Specific Sphingosine-1-Phosphate Receptors Augments a Repair Mediating Schwann Cell Phenotype"

_ijms, 2022, doi:10.3390/ijms231810311_

Round 1

Reviewer 1 Report

The manuscript by Schira-Heinen et al. documents: i. the expression pattern of sphingosine-1-phosphate receptors (S1P1-5) in cultured rat neonatal Schwann cells (SC), ii. the role of the particular S1P in creating the phenotype of SC. Current knowledge on this topic is scarce, therefore the results of this paper are novel and interesting.

The main concern are following:

1. In some of the experiments the Authors compared the amount of the proteins between the particular samples. To do that, they loaded equal amounts of total proteins (30 or 50 micrograms) from each sample in each line. However, according to the manuscript, they detected the proteins of interests, but they did not use any house-keeping protein detection or staining of the total protein transferred on the membrane. Given that multiple steps are between the protein determination and the visualization of blotted proteins (with a subsequent densitometric analysis), several factors can contribute to unequal amounts of proteins at the membrane level. Why the detection of the house-keeping protein or staining of the proteins transferred on the membrane was not performed? Without one of those it is difficult to say whether the observed changes in the amount of the proteins of interest result from the modulation performed in this study or from one of several factors disturbing the intended equal amounts of proteins.

2. Most of the results show the gene expression, not the protein level. Are the Authors sure that the changes observed at mRNA level translate into the changes at the protein level? It would be reasonable to examine the protein level, at least in the cases where mRNA is changed. This could particularly interesting in case of S1P receptors, since as the Authors mentioned, data on the level of S1P proteins in SC are limited and inconsistent.  

Round 2

Reviewer 2 Report

I thank the authors for their comprehensive answers to my questions.